# Enhancing Liver Transplant Outcomes through Liver Precooling to Mitigate Inflammatory Response and Protect Mitochondrial Function

**DOI:** 10.3390/biomedicines12071475

**Published:** 2024-07-04

**Authors:** Minh H. Tran, Jie Gao, Xinzhe Wang, Ruisheng Liu, Colby L. Parris, Carlos Esquivel, Yingxiang Fan, Lei Wang

**Affiliations:** 1Department of Molecular Pharmacology and Physiology, University of South Florida College of Medicine, Tampa, FL 33612, USA; 2School of Health Professions, The University of Alabama at Birmingham, Birmingham, AL 35294, USA

**Keywords:** hypothermia, mitochondria, liver transplantation, graft function

## Abstract

Transplanted organs experience several episodes of ischemia and ischemia-reperfusion. The graft injury resulting from ischemia-reperfusion (IRI) remains a significant obstacle to the successful survival of transplanted grafts. Temperature significantly influences cellular metabolic rates because biochemical reactions are highly sensitive to temperature changes. Consequently, lowering the temperature could reduce the degradative reactions triggered by ischemia. In mitigating IRI in liver grafts, the potential protective effect of localized hypothermia on the liver prior to blood flow obstruction has yet to be explored. In this study, we applied local hypothermia to mouse donor livers for a specific duration before stopping blood flow to liver lobes, a procedure called “liver precooling”. Mouse donor liver temperature in control groups was controlled at 37 °C. Subsequently, the liver donors were preserved in cold University of Wisconsin solution for various durations followed by orthotopic liver transplantation. Liver graft injury, function and inflammation were assessed at 1 and 2 days post-transplantation. Liver precooling exhibited a significant improvement in graft function, revealing more than a 47% decrease in plasma aspartate transaminase (AST) and alanine aminotransferase (ALT) levels, coupled with a remarkable reduction of approximately 50% in liver graft histological damage compared to the control group. The protective effects of liver precooling were associated with the preservation of mitochondrial function, a substantial reduction in hepatocyte cell death, and a significantly attenuated inflammatory response. Taken together, reducing the cellular metabolism and enzymatic activity to a minimum level before ischemia protects against IRI during transplantation.

## 1. Introduction

Liver transplantation is a life-saving procedure and is often the only treatment option available for patients with end-stage liver disease and acute liver failure [1,2,3,4]. A significant limitation of liver transplantation is the unavoidable hepatic ischemia-reperfusion injury (IRI), which poses a critical risk factor for the survival of transplanted grafts [5,6,7,8]. Several episodes of ischemia occur during transplantation, including warm ischemia during organ procurement, cold ischemia during the preservation period, and warm ischemia again during implantation. The subsequent reperfusion phase, beginning when blood flow and oxygen levels in the liver are restored, exacerbates the initial damage. The injury induced by IR primarily occurs due to the prolonged metabolic demands imposed on a warm ischemic organ and the insufficient supply of oxygen to fulfill these requirements.

One approach to enhance the liver’s tolerance to ischemia is lowering the organ’s temperature [9]. A 50% reduction in cellular metabolic rate occurs with every 10 °C decrease in temperature according to the Arrhenius equation. Building on this principle, the invention of cold preservation solution in the late 1960s allowed successful storage of organs in a simple hypothermia solution for up to 15 h [10,11,12,13]. Recently, machine perfusion (MP) at different temperatures has been tried on DCD liver grafts; however, the application of MP to DCD liver transplantation has remained challenging [14,15]. The optimal parameters for the use of MP in DCD livers have not been well established [12,16]. Numerous experimental and clinical studies have been conducted and various drug products have been tested in clinical trials in the peri-transplant period. While some have demonstrated early beneficial effects on IRI, their benefit to long-term graft is yet to be fully demonstrated [17,18,19,20,21,22]. The pathophysiological changes associated with IRI in liver transplantation are not yet well defined, although they have been studied extensively. Future double-blinded, randomized, large-scale clinical trials will be necessary to assess the advantages of MP technology.

Temperature significantly influences cellular metabolic rates as biochemical reactions are greatly sensitive to temperature variations [23,24]. Therefore, it seems quite likely that most ischemia-triggered degradation reactions can be lessened by lowering tissue temperature [25,26,27]. Therapeutic hypothermia has been used in surgeries, including liver surgery, since the 1960s [28,29,30]; however, its significance is still controversial and not conclusive [31,32,33,34,35]. Whether inducing local hypothermia in the liver before the stopping of blood flow protects against IRI has not been explored. In the present study, we demonstrated that reducing the cellular metabolism and oxygen consumption in the liver to a minimum level by local hypothermia before ischemia ameliorates IRI during transplantation.

## 2. Materials and Methods

### 2.1. Animals

The use of animals in the experiments was conducted following the approved guidelines from the Institutional Animal Care and Use Committee at the University of South Florida and the National Institutes of Health’s Guide for the Care and Use of Laboratory Animals. Male C57BL/6J mice, aged 12 weeks with a body weight of about 28–32 g, were obtained from Jackson Lab (Indianapolis, IN, USA). After arrival, the animals were housed in a temperature-controlled environment with 12:12 h light–dark cycle and ad libitum access to water and food for 1 week before the experiments. The animals were randomly divided into control or precooling groups and donors or recipients based on the requirements of the experiments. All chemicals were purchased from Sigma-Aldrich (St. Louis, MO, USA) unless otherwise indicated. Animals were euthanized as needed according to the guidelines set forth by the American Veterinary Medicine Association.

#### 2.1.1. Liver Transplantation with or without Precooling

Orthotopic liver transplantation (OLT) was performed in mice of the same sex, employing the cuff technique as described in the literature [36]. Briefly, donor mice were anesthetized using isoflurane, and the body temperature was kept at 37 °C. A midline abdominal incision was performed to expose the infra hepatic vena cava (IHVC) and supra hepatic vena cava (SHVC) from the left diaphragmatic vein, right suprarenal vein, and right renal vein. A 22G stent was then inserted into the bile duct, followed by exposing the portal vein (PV) from the pyloric vein. A polystyrene plate was used to isolate the liver lobes from the animal body. The isolated liver lobes were then surrounded with 4 cm length Penrose tubing (3/4″ Diameter) filled with 4° (precooling) or 37 °C (control) water and circulated with a pump for 30 min. Liver tissue temperature changes were monitored with a flexible probe (Homeothermic Monitoring System, Harvard Apparatus from Holliston, MA, USA) placed between the median and left lateral lobes during precooling. The liver was then flushed with 5 mL of cold University of Wisconsin (UW) solution via PV and removed. Lastly, the obtained graft was preserved in 4 °C UW solution for 6 h. The cuff was prepared before implantation as previously described [36]. 

The recipient mice were anesthetized, and a midline abdominal incision was performed. The anastomosis of SHVC was performed with a running 10-0 suture after removing the original liver. During the recipient operation, two-cuff anastomosis technology was utilized to restore the blood flow of the PV and the IHVC. A biliary stent was used to connect the bile duct. Finally, the abdomen was sutured shut, and the mice were given time to recover.

The sham group also underwent similar procedures and time courses except for the liver transplantation.

#### 2.1.2. Liver Graft Injury and Oxidative Stress Biomarkers

The function of the liver graft was assessed through the measurement of plasma aspartate transaminase (AST), alanine aminotransferase (ALT), and total bilirubin levels by Antech Diagnostics at 1 and 2 days following transplantation. Oxidative stress was evaluated by measuring the liver malondialdehyde (MDA) and the carbonyl content levels, as described previously [37,38]. Briefly, liver tissue was homogenized and incubated with 2,4-dinitrophenylhydrazine. The carbonyl content was determined by utilizing a spectrophotometer set at 370 nm, relying on the reaction between carbonyl groups and 2,4-dinitrophenylhydrazine to produce a 2,4-dinitrophenylhydrazone. Liver MDA levels were assessed using high-performance liquid chromatography at 250 nm with an LC-18 DB column [37].

### 2.2. Histopathological Examinations

At the end of experiment, the liver specimens were fixed in 4% paraformaldehyde and embedded in paraffin. The liver samples were sectioned into 4 μm slices and stained with hematoxylin and eosin (H&E). The severity of liver injury was graded from 0 to 4 for sinusoidal congestion, vacuolization of the hepatocyte cyto-plasm, and parenchymal necrosis, as described by Suzuki et al. [39]. For statistical analysis, at least 5 visual fields of each specimen were randomly selected and photographed under the microscope (200×, Olympus BX53, Olympus American INC., Breinigsville, PA, USA), and analyzed by Fiji/ImageJ V2.

Hepatocyte cell death was assessed by Terminal dUTP Nick-End Labeling (TUNEL) staining with the in situ Cell Death Detection Kit (Roche, catalog no. 11684795910) according to the manufacturer’s instructions. TUNEL-positive cell/nuclei were quantified as the percentage (%) of TUNEL and DAPI double-positive cells relative to total cells (DAPI-positive cells). For statistical analysis, 5 visual fields of each specimen were randomly selected and photographed with a fluorescence microscope (Keyence BZ-X710, KEYENCE CORPORATION OF AMERICA, Itasca, IL, USA) and analyzed with Fiji/ImageJ V2.

Immunohistochemical staining (IHC) was performed as previously described [40]. Briefly, the liver slices were subjected to antigen retrieval with a sodium citrate buffer and permeabilized with 0.3% Triton-X 100 in PBS, blocked with 5% normal goat serum, and probed with the primary antibody Caspase 3 (Abcam, ab184787, Waltham, MA, USA) at the dilution of 1:1000 overnight followed by incubation with Goat anti-Rabbit lgG H&L (HRP) (Abcam, ab97051). The percentages of Caspase 3 positive cells were analyzed with Fiji/ImageJ.

All morphometric analyses were performed blinded to the experimental procedures.

### 2.3. Mitochondria Isolation and Function Evaluation 

One separate group of recipients were euthanized 1 day after transplantation for liver mitochondria activity and function evaluation. Mitochondria were isolated and the complex activities and the oxygen consumption rates were measured with a Seahorse XFe24 Analyzer (Agilent Technologies, Inc., Santa Clara, CA, USA) as previously described [41]. Fresh liver samples of 400 mg were washed with 4 °C isolation buffer (210 mM mannitol, 70 mM sucrose, 5 mM HEPES, 1 mM EGTA, 0.5% BSA pH 7.2) and homogenized using a glass Dounce Homogenizer. The homogenate was centrifuged at 1000× *g* for 5 min at 4 °C. The supernatant, devoid of fat, was gathered and centrifuged twice at 12,000× *g* for 10 min at 4 °C to yield a mitochondrial pellet, which was subsequently resuspended in a small volume of isolation buffer. The mitochondrial protein concentration was quantified using a BCA protein assay kit (Pierce, 23227, Rockford, IL, USA).

Mitochondrial Complex I Activity Mitochondrial OXPHOS complex I (NADH dehydrogenase) enzyme activity was assessed with a Complex I Enzyme Activity Microplate Assay Kit (Colorimetric) (ab109721; Abcam) following the manufacturer’s instructions. Briefly, isolated mitochondria samples with a concentration of 5.5 mg/mL were loaded into the wells of the microplate and incubated for 3 h at room temperature. Then, 200 μL of assay solution was added to each well, and the optical density (OD) at 450 nm was monitored using a microplate reader for 30 min at room temperature.

Mitochondrial oxygen consumption rates (OCR) were measured with a Seahorse XFe24 Analyzer (Agilent Technologies, Inc.), as previously described [41,42]. In summary, 2 µg of freshly isolated mitochondria was introduced into a mitochondrial assay solution (MAS), which included 70 mM sucrose, 220 mM mannitol, 10 mM KH_2_PO_4_, 5 mM MgCl_2_, 2 mM HEPES, 1 mM EGTA, and 0.2% BSA in nanopure water, with the pH adjusted to 7.4 using KOH. Additionally, the solution contained 10 mM succinate and 2 µM rotenone. This mixture was then placed into 24-well Seahorse plates. The respiratory reagents were sequentially loaded into the drug ports of a hydrated sensor cartridge as follows: (A) ADP at a final concentration of 1 mM, (B) oligomycin at a final concentration of 2 µM, (C) carbonyl-cyanide-4-(trifluoromethoxy) phenylhydrazone (FCCP) at a final concentration of 4 µM, and (D) rotenone and antimycin A, each at a final concentration of 2 µM. A minimum of three cycles of OCR measurements were conducted in the respiration assay. The basal respiration (state 2), phosphorylating respiration in the presence of ADP (state 3), resting respirations with oligomycin (state 4 or state 4o, the ATP synthase is inhibited with oligomycin), maximal uncoupling respiration in the presence of FCCP (state 3u), and the response to antimycin A were determined. The results were analyzed using the Wave software (Agilent Technologies, Inc., https://www.agilent.com/en/product/cell-analysis/real-time-cell-metabolic-analysis/xf-software/seahorse-wave-desktop-software-740897, accessed on 1 July 2024) export option and GraphPad Prism.

### 2.4. Cytokine Profiles of Liver Lysates

At the end of experiment, liver tissue sections were harvested, homogenized, and treated with protease/phosphatase inhibitor cocktail (5872, Cell Signaling Technology, Danvers, MA, USA). The levels of cytokines were evaluated by the Proteome Profiler Mouse XL Cytokine Array (ARY028, R&D Systems, Boston, MA, USA) for 111 different analytes involved in the inflammatory response, the according to manufacturer’s instructions. Array panels were visualized and captured with a ChemiDoc Imager (Bio-Rad, Inc., Hercules, CA, USA). Pixel densities of the cytokine blot spots were analyzed using ImageJ V2 software (National Institutes of Health, Bethesda, MD, USA) and normalized using reference spots.

### 2.5. Real-Time PCR

Total RNA isolation and real-time PCR were performed as previously described [43]. Briefly, total RNA was extracted from fresh liver tissue using Trizol reagent (Invitrogen, Carlsbad, CA, USA) and converted to cDNA using the High-Capacity cDNA Reverse Transcription Kit (Applied Biosystems, Foster City, CA, USA). The primer pairs used for amplification of the genes of interest are described in Table 1. β-actin was the invariant control in the experiment. Quantitative PCR analysis was then performed using CFX96 Real-Time Detection System (Chromo4, Bio-Rad, CA, USA) and iQ SYBR Green Supermix (iTaq SYRB, Bio-Rad, CA, USA), according to the protocol of the manufacturer. The amount of each cDNA relative to the β-actin was determined using the 2^−ΔΔCt^ method.

### 2.6. Statistics

Experimental values are presented as mean ± SEM unless otherwise indicated in the figure legends. Statistical analysis was performed using Prism 10 (GraphPad Software; San Diego, CA, USA) or SigmaPlot13.0 software. Statistical tests for each dataset are specified in the figure legends, where statistical significance is defined as *p* < 0.05. Comparisons of the mitochondria bioenergetic analysis and liver function were performed by a two-way analysis of variance (ANOVA) followed by a Tukey multiple comparison test. Comparisons of the datasets of histology and RT-PCR were performed using Student’s *t* test.

## 3. Results

### 3.1. Liver Temperature Changes during Precooling

The experimental procedure is depicted in Figure 1. During precooling, the core of the liver lobes and body temperatures were monitored with a Signals 2-Channel Alarm Thermometer assembled with two microprobes. One microprobe was positioned under the median lobe and between the left lateral and the right lateral lobes, and the second one was used to measure the body temperature, which was maintained at 36.8–37 °C utilizing an auxiliary heating lamp and warm water circulating blankets. The liver temperature dropped from 37 °C to about 8 °C within 8 min and remained at 6–8 °C for 30 min. Then the liver was flushed with cold UW solution. The temperature of the liver immediately decreased to about 4 °C. For the livers without precooling, the temperature was maintained at 36.8–37 °C before liver flushing and gradually decreased to about 13 °C after flushing (Figure 2, n = 5/group). Variations in temperature alterations exerted a notable influence on oxygen consumption and ATP preservation, stemming from the distinct declines in metabolic rates.

Male C57BL/6J mice were randomly divided into the groups as described in the methods. For the donor in control group (A), the liver lobes were surrounded with 37 °C water circulation for 30 min, followed by cold storage and OLT and liver graft function evaluation. For the donor in the precooling group (B), the liver lobes were surrounded with 4 °C water circulation for 30 min, followed by cold storage and OLT and liver graft function evaluation (n = 5).

For the precooling group, the liver temperature dropped from 37 °C to about 8 °C within 8 min, remained at 6–8 °C for 30 min and quickly decreased to about 4 °C upon flushing with cold saline. In the livers without precooling, the temperature was maintained at 36.8–37 °C and gradually decreased to about 13 °C after flushing.

### 3.2. Precooling Preserved Mitochondria Activities and Function

Mitochondria are the primary source of ATP generation through oxidative phosphorylation. Mitochondrial metabolism is sensitive to temperature. In cold ischemia, ATP depletion leads to an increase in the osmotic gradient, causing mitochondrial matrix swelling, which in turn inhibits mitochondrial respiration [44].

To investigate the impact of precooling on mitochondria function preservation by slowing the mitochondria metabolism rate and delaying ATP depletion, we isolated mitochondria from liver tissue post-transplantation through sequential centrifugations. We first tested the influence of precooling on the activities of Complex I in the isolated liver mitochondria. Figure 3A illustrates that OLT significantly decreased the activity of Complex I in the control group’s isolated liver mitochondria by 47% compared to the sham group. Notably, precooling demonstrated a substantial preservation of Complex I activity in the recipient liver mitochondria, indicated by approximately 50% higher activity than the control group (n = 5/group).

Next, we aimed to confirm their consequent occurrence in the mitochondria function by measuring the OCR in isolated mitochondria of both the precooling and control groups using the Seahorse XFe24 Analyzer. As shown in Figure 3B,C, basal OCR was significantly higher in the precooling group than the control group (n = 5/group). State 3 phosphorylating respiration induced by addition of ADP was about 30% greater in the precooling group than that in the control group. There was a minor difference observed between the control and precooling groups in state 4o respiration, which was not statistically significant. FCCP (state 3u) increased OCR for both groups, but the increase was greater in the precooling group than that in the control group. These studies demonstrated that liver precooling attenuated the overall mitochondrial damage, albeit with an unremarkable preservation of mitochondrial function.

### 3.3. Liver Precooling Protected Transplanted Graft Function against IRI

Liver precooling was applied 30 min before the liver procurement to the C57BL/6 donor mice, followed by cold storage for 6 h. Liver transplantation was then performed. Graft functions were evaluated by measurement of plasma AST, ALT, and total bilirubin levels by Antech Diagnostics at 1- and 2-days following transplantation. All recipients were tested in similar basal liver chemistry before transplantation. Recipients from donors with liver precooling (n = 5) exhibited significantly lower liver enzyme levels. The plasma ALT and AST were 42–46% lower in the precooling group than the control group as shown in Figure 4A,B. Total bilirubin was about 53% lower in the precooling groups than the control group (Figure 4C). These data indicate that precooling pretreatment protects against liver IRI and preserved graft hepatocellular function.

### 3.4. Liver Precooling Attenuated Liver Cell Death

Histopathology analysis by H&E staining revealed more severe sinusoidal congestion, vacuolization, and hepatocytes necrosis in the control group than in the precooling group at 2 days after OLT. TUNEL staining showed 43% less cell death in the precooling group than control group, which was consistent with the serum ALT levels and H&E staining (Figure 4D). The Caspase-cascade system plays a crucial role in liver IRI. Apoptotic active Caspases-3 directly caused hepatocellular apoptosis and reflected the progress of apoptosis in the ischemic liver graft. Next, we assessed the expression of cleaved Caspases-3 by immunohistochemistry analysis. Along with the TUNEL assay, the activity of Caspases-3 was significantly repressed in the liver graft tissue in the precooling group compared with the control group.

The enhanced functional parameters observed correlated with a reduction in histopathological evidence of injury. This indicates that the implementation of precooling effectively shielded the donor liver from ischemic injury during cold storage and consequently improved the function of the transplanted graft.

### 3.5. Liver Precooling Suppressed Oxidative Stress and Inflammatory Response

During liver transplantation, IRI triggers a complex inflammatory program involving cytokines and chemokines. Following liver IRI, inflammatory cytokines and chemokines stimulate the release of reactive oxygen species (ROS), cytokines, myeloperoxidase (MPO), and various mediators, amplifying tissue damage. To evaluate the hepatoprotective effects of precooling in the context of oxidative stress and inflammatory stimulation, we compared oxidative damage parameters and cytokine profiles in liver lysates between the precooling and control groups.

The cytokine array revealed that the precooling treatment led to a decreased expression of more than 20 cytokines and inflammatory mediators. The heat map illustrates cytokines and inflammatory mediators with a reduction in expression of more than 2-fold compared to the control group (Figure 5A,B), including CCL6/C10, Chitinase 3-like 1, MMP9, and Pentraxin 2/SAP. Subsequently, we conducted RT-PCR to assess the mRNA levels of these and inflammatory mediators. Precooling of the liver led to a substantial decrease, ranging from 30% to 80% in the mRNA expression of CCL6/C10, Chitinase 3-like 1, MMP9, and Pentraxin 2/SAP in the precooling group compared to the non-precooling group (Figure 5C). Liver precooling significantly reduced the oxidative damage parameters assessed in this study. Specifically, liver carbonyl levels, indicative of protein oxidative damage, were found to be significantly lower in the precooling group than the control group post-transplantation. While both the precooling and control groups showed a significant increase in total MDA levels, the magnitude of this increase was notably lower in the precooling group compared to the control group (Figure 5D). This suppression of the oxidative stress and inflammatory response, alongside a reduction in metabolic rate and oxygen consumption, contributes to the protective effects of precooling on liver grafts, thereby mitigating ischemic injury during liver transplantation.

## 4. Discussion

Overcoming IRI and preserving organ quality are crucial challenges in transplantation surgery. Ongoing studies aim to find new methodologies for reducing liver IRI as there are currently no clinically approved strategies to completely protect the liver from IRI. In this study, we explored precooling as a drug-free approach with profound protective effects against IRI during liver transplantation.

The protective effect of hypothermia is linked to a decrease in liver metabolism, leading to a subsequent reduction in oxygen requirement as temperature decreases. Prior research has established the protective effects of mild hypothermia against neurological impairment, cardiac arrest, ischemic stroke, and kidney damage [30,45,46]. However, these studies typically involved systemic application, limiting temperature reduction due to potential adverse effects on the overall system such as hypotension, cardiac arrhythmias, and metabolic acidosis [47]. Selective liver hypothermia or topical liver cooling avoids the systemic side effects and has been proven to be effective in some clinical trials [48,49]. Nevertheless, in all these reports, the utilization of hypothermia during organ procurement or implantation was emphasized, while the warm ischemic period occurring after vascular clamping and before organ perfusion was overlooked. In our ongoing research, we have implemented localized and topical precooling to specifically tackle this overlooked aspect. By circulating 4 °C water around the liver lobes, we achieved continuous liver temperature reduction without affecting core temperature. This method rapidly lowered the liver temperature to 6–8 °C, reducing the metabolic rate of the liver cells before vasculature clamping and surpassing the reduction achieved with conventional systemic mild hypothermia. Consequently, it enabled a more significant decrease in the oxygen requirement.

Precooling is expected to not only lower the energy demand but also diminish the accumulation of superoxide radicals, delaying ATP decline and consequently mitigating oxidative stress during ischemia. Elevated ROS production during liver ischemia-reperfusion triggers an oxidative stress response, leading to cellular damage and apoptosis [50]. Mitochondria are identified as the primary source of ROS products [51]. Notably, moderate hypothermia has demonstrated protective effects on isolated perfused rat livers by mitigating oxidative stress [52,53]. The reactive oxygen metabolite levels were observed to decrease during the hypothermic stage of treatment and returned to control-group levels after rewarming. In our current investigation, we observed that levels of liver MDA and carbonyls remained significantly lower in the precooling group compared to the control group post-transplantation. This observation suggests that while ROS generation may return to basal levels after rewarming, the activation of antioxidant defenses may impede protein damage and lipid peroxidation cascades.

The reduced oxidative damage may be correlated with the preserved mitochondrial respiratory enzyme activity. Studies have shown that mitochondrial ROS production exhibits an inverse relationship with oxidative phosphorylation rates [54,55]. Our results revealed a notably higher activity of Complex I within the precooling group than in the control group, suggesting that precooling likely mitigates the impairment of respiratory enzymes induced by IRI. The enhanced function of mitochondrial respiratory enzymes may play a pivotal role in sustaining the integrity of the mitochondrial respiratory chain, preserving membrane permeability, regulating Ca^2+^ levels, and fortifying mitochondrial defense mechanisms. Our high resolution respirometry data obtained with the Seahorse XF Analyzer confirmed that precooling preserved mitochondria function and alleviated IR-induced mitochondrial injury in livers.

Liver contains all the subtypes of lymphocytes as resident cells [56,57,58,59], acting as the source for the production of cytokines and chemokines [56,60,61]. These molecules, in turn, activate neutrophils during the late phase of reperfusion injury [62,63], and induce the release of ROS, cytokines, myeloperoxidase and various other mediators, all of which amplify the tissue damage [63,64,65,66]. It is widely recognized that an inflammatory response, dependent on neutrophils, contributes to IRI. Kato et al. demonstrated that hypothermia, with temperatures as low as 25 °C, attenuates the hepatic inflammatory response induced by IRI [67]. In this study, we investigated if local precooling could blunt the inflammatory response following transplantation by using the Proteome Profiler Mouse XL Cytokine Array kit. We identified more than 20 inflammatory mediators involved in the inflammatory response that were significantly suppressed by the precooling. Further investigations will be necessary to discern the specific mediators within the inflammatory cascade that are modulated by precooling.

Our current study provides pre-clinical evidence that application of local precooling can protect against IRI-induced acute liver injury. Its application in the mouse model of liver transplantation revealed protection of liver graft function, which was associated with attenuated mitochondrial oxidative stress, suppressed inflammatory response and subsequently decreased hepatocytes necrosis, fibrosis and apoptosis.

There are some limitations in our work. First, we only aimed to assess the effects of precooling on short-term graft function rather than long-term function. The detailed mechanisms of the antioxidant enzymes, which contribute to oxidative stress in mitochondria and cytokines, will be investigated in the continuing study. Further research with larger animal models is necessary to validate the findings presented here. In summary, our study describes a new strategy of local precooling that attenuated oxidation stress and inflammatory response in the liver, ameliorated hepato-cellular damage and improved transplanted liver graft function. This technique has potential not only for the improvement of graft survival, but also for the expansion of the donor pool.

## Figures and Tables

**Figure 1 biomedicines-12-01475-f001:**
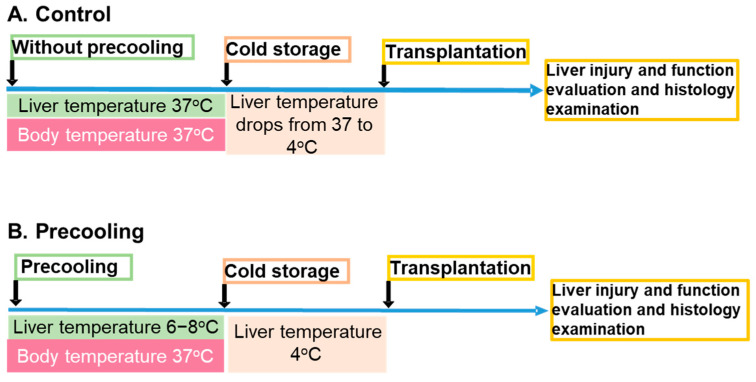
Experimental procedure.

**Figure 2 biomedicines-12-01475-f002:**
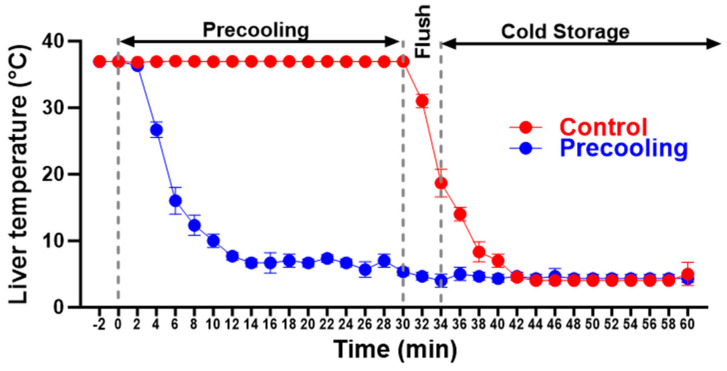
Liver temperature changes during the precooling.

**Figure 3 biomedicines-12-01475-f003:**
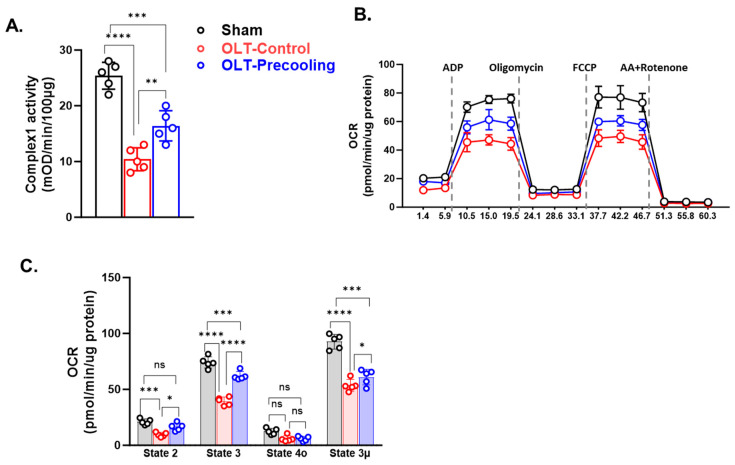
Liver mitochondrial function is preserved with the application of precooling. (**A**) Mitochondria Complex I activities were preserved in the precooling group. (**B**) OCR traces of isolated mitochondria from liver grafts with or without precooling, expressed as picomoles of O2 per minute, under basal conditions and after the injection of adenosine diphosphate (ADP; 1 mM), oligomycin (Oligo; 2 µM), carbonyl-cyanide-4-(trifluoromethoxy) phenylhydrazone (FCCP; 4 µM), and antimycin A (AA) + rotenone (2 µM). (**C**) Analysis of mitochondrial respiratory parameters obtained from normalized XFe24 graphs (**B**). Data are presented as means ± SEM. Statistical significance was assessed by one-way ANOVA followed by Tukey multiple comparisons test (* *p* < 0.05, ** *p* < 0.01, *** *p* < 0.001, and **** *p* < 0.0001; ns, not significant; n = 5 mice per group).

**Figure 4 biomedicines-12-01475-f004:**
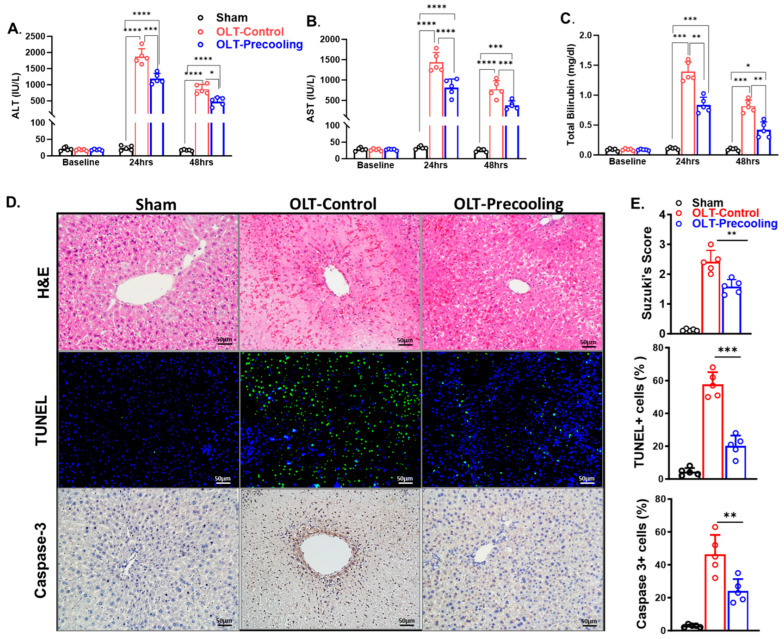
IRI–induced liver graft injury is attenuated by the application of precooling. Plasma ALT (**A**), AST (**B**) and total bilirubin (**C**) levels measurements on day 1 and 2 after transplant demonstrated decreased liver injury by the application of precooling. (* *p* < 0.05, ** *p* < 0.01, *** *p* < 0.001 and **** *p* < 0.0001, Statistical significance was assessed by two-way ANOVA followed by Tukey multiple comparisons test. n = 5 mouse recipients.) (**D**) Representative liver tissue sections stained with H&E, TUNEL, and immunofluorescent anti-Caspase-3. (**E**) Quantification of sinusoidal congestion, vacuolization and hepatocytes necrosis area over whole-liver sections, TUNEL-positive cells/nuclei as the percentage (%) of TUNEL, and the percentage of Caspase 3 positive cells/nuclei. Scale bar is 50 µm. Data are presented as means ± SEM. One-way ANOVA followed by Tukey multiple comparisons test ** *p* < 0.01, *** *p* < 0.001, n = 5 liver sections per group).

**Figure 5 biomedicines-12-01475-f005:**
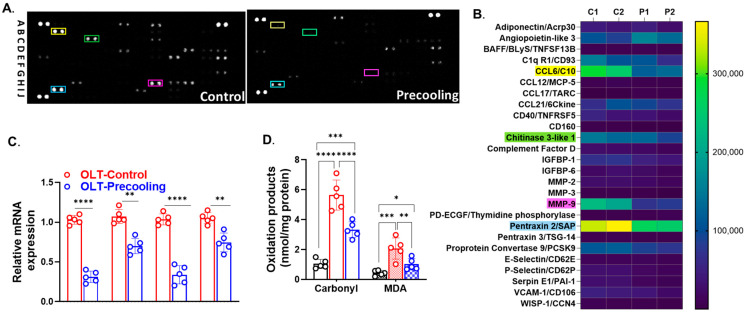
Liver precooling suppressed inflammatory response and oxidative stress. (**A**) Cytokine Arrays detection of multiple analytes in mouse liver graft tissues of control and precooling groups. Each blot represents immunoreactive labeling against the respective antibodies. The blots labeled with colored squares represent cytokines that exhibited significant differences between the two groups. (**B**) Heatmap representation of the quantification of the cytokine profile of control (C1, and C2) and precooling groups (P1 and P2). Colored squares highlight the cytokines most depressed by precooling, (n = 2 biologically independent samples/group). (**C**) RT-PCR confirmed the mRNA levels of the genes exhibiting the most significant differences between the control and precooling groups. (** *p* < 0.01 and **** *p* < 0.0001; statistical significance was assessed by One-way ANOVA followed by Tukey multiple comparisons test; n = 5 mouse recipients). (**D**) Liver carbonyl and MDA levels demonstrated reduced oxidative damage in the precooling group than control group. (* *p* < 0.05, ** *p* < 0.01, *** *p* < 0.001 and **** *p* < 0.0001; statistical significance was assessed by two-way ANOVA followed by Tukey multiple comparisons test; n = 5 mouse recipients).

**Table 1 biomedicines-12-01475-t001:** Primer pairs for the RT-PCR.

Genes	Primers
MMP9	Forward 5′-GCTCATGTACCCGCTGTATAGCT-3′
	Reverse 5′-CAGATACTGGATGCCGTCTATGTC-3′
CCL6	Forward: 5′ ATGAGAAACTCCAAGACTGCC 3′
	Reverse: 5′ TTATTGGAGGGTTATAGCGACG 3′
β-actin	Forward 5′ CCGGGACCTGACAGACTA 3′
	Reverse 5′ AGAGCCTCAGGGCATCGGAAC 3′
CHI3L1	Forward 5′CCCAACCTGAAGACTCTCTTG
	Reverse 5′ CCAAGATAGCCTCCAACACC

## Data Availability

All the data and materials supporting the findings of this study are available within the article. Further enquiries can be directed to the corresponding author.

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
