# Peer review of "Enhancing Liver Transplant Outcomes through Liver Precooling to Mitigate Inflammatory Response and Protect Mitochondrial Function"

_biomedicines, 2024, doi:10.3390/biomedicines12071475_

Round 1

Reviewer 1 Report

Comments and Suggestions for Authors

You describe a strategy of local pre-cooling that improved transplanted liver graft function in this rodent model. Retrospectively you assessed a small number of patients. There is emerging data in human livers that stem cells may be advantageous  in a variety of ways. In your centers liver transplant patients any studies?

Author Response

We sincerely appreciate all valuable comments and suggestions from the reviewers and editors, which helped us improve the quality of our manuscript. We have thoroughly revised the manuscript as the reviewers suggested. We believe that all the reviewers’ and editor’s concerns have been carefully addressed. Our point-by-point response to the reviewers' comments is provided below in blue.

Review 1. You describe a strategy of local pre-cooling that improved transplanted liver graft function in this rodent model. Retrospectively you assessed a small number of patients. There is emerging data in human livers that stem cells may be advantageous in a variety of ways. In your centers liver transplant patients any studies?

Response: Thank you for pointing out this issue. Stem cell therapy in organ transplantation has been investigated for decades. Stem cells possess many properties, including immunomodulation, differentiation into hepatocytes, and repair of damaged tissues. Currently, stem cells have been preliminarily proven to be effective in the short term. However, the long-term efficacy and safety, as well as the regulatory mechanisms of stem cell development and differentiation, have not been established. Many challenges and problems remain to be solved, including the identification and differentiation of stem cells, generating functional and transplantable hepatocytes on a large scale, reducing immunogenicity, and addressing the progression of liver disease and hepatocellular carcinoma. Tampa General Hospital has a Cell Therapies unit for treating patients with complex cancers, but it has not been used in liver transplantation.

Review 2. The authors set out to investigate the possibility of improving liver transplantation by precooling, clearly realizing that this method is more than 50 years old and should be done something novel. Still, there are inconsistencies in the quality of grafts taken by two different techniques. The work is well done and may interest readers of Biomedicine. However, there are a few points that need to be corrected.

Response: Thank you for your insightful feedback and for recognizing the potential impact of our work. We are grateful for your positive assessment of our work and believe that the corrections and clarifications we made below and incorporated into the main manuscript will further strengthen the manuscript. We hope that the improved manuscript will meet the standards of Biomedicine's readership.

Taking into account the authors' phrase..... However, outcomes from DCD transplants have historically shown a relative inferiority when compared to liver transplantation from a living donor or donors after brain death [9,10]… it should be clearly stated that the authors are modeling this very situation (namely DCD, don’t they?).

Response: Thank you for pointing this out. We have deleted this sentence from the manuscript as we did not include the DCD in our study.

….rotenone + antimycin A (2 μM final)….. 2 μM final of what? Rot? Or AntA? Or both by 2 μM?

Response: We meant that the final concentrations for both rotenone and antimycin are 2µM. We have clarified this in the main manuscript as “rotenone + antimycin A, each at a final concentration of 2 µM.”

The authors show that the activity of complex 1 drops by a factor of 2.5 (from 25 to 10). At the same time in SeaHorse experiments, this should be reflected on the starting point, and there the difference is from about 12 to 20, i.e. the difference is significantly less than 2-fold. The authors should, first of all, say what respiratory substrate they used and somehow explain this difference, which is small, but somewhere there is a reason for it. Moreover, if we calculate the respiratory control, by eye ball it seems that all three curves show the same respiratory control, i.e. the oxidative phosphorylation system is not damaged, but simply the number of working mitochondria decreases.  It also should be written somewhere and I wonder how it could be since oxidative phosphorylation is the most sensitive to damage factor.

Response: Thank you for raising this important question. We have clarified the substance used in the OCR measurement in the method section and here: Two micrograms of freshly prepared mitochondria in mitochondrial assay solution (MAS) (70 mM sucrose, 220 mM mannitol, 10 mM KH2PO4, 5 mM MgCl2, 2 mM HEPES, 1 mM EGTA, and 0.2% BSA in nanopure water, pH 7.4, adjusted using KOH) were loaded onto the 24-well SeaHorse plates. Succinate (10 mM) and Rotenone (2 μM) were added to the MAS and the resulting solution was used to make 10 × stocks of the respiratory inhibitors and uncoupler. The substance of Succ/Rot (succinate with rotenone) bypasses complex I to feed electrons directly to the ubiquinone pool. There are no significant differences between the precooling and sham groups in state 2 before the addition of ADP, but significant differences exist between the precooling and control groups (Fig. 3C). This demonstrates that oxidative phosphorylation is more damaged in the control group than in the precooling group.

Fig 3 C. What is state 4o? (Maybe state 4?). What is state 3μ? (Maybe state 3u?)

Response: State 4o means that the ATP synthase is inhibited with the addition of oligomycin (State 4o). We added the explanation in the main text. We apologize for the typo in "state3µ." We have corrected it in the main text to "state 3u."

Review 3.

This is an interesting study. The reviewer has some comments as follows:

  1. This manuscript is well written. After searching the literature, this study has its novelty.

Response: Thank you for your positive feedback and for recognizing the novelty of our study.

  1. Basically, the experimental design of the animal model in this study is reasonable.

Response: We really appreciate your positive evaluation of our experimental design.

  1. In the Methods section, the fonts of the subtitles are inconsistent, please check the fonts.

Response: Thank the reviewer for pointing this out. We have carefully reviewed and standardized all the fonts.

  1. In line 277 and Figure 3C, please check the words “state 3μ” (state 3u).

Response: we apologize for the typo. We have corrected it to state 3u.

  1. In Figure 4D, please check the scale of image for caspase-3 IHC in OLT-Control group. The cells in this image appear to be much smaller than those in the sham control group.

Response: We have replaced the image with another from the same group and confirmed the scales.

  1. In Table 1, please confirm whether the font is consistent with the manuscript.

Response: We have updated the fonts and ensured they are consistent with the rest of the manuscript.

  1. In Figure 6, the data presentation for donor body temperature during liver procurement surgery correlates with transplanted graft outcomes is confusing. The conditions of these patient data are very different from the animal data. The temperature changes only 34-38oC and there are no precooling or cold processes in patient data. These patient data cannot echo the results of animal experiments. Therefore, please consider whether the presentation of these patient data is necessary.

Response: Thank you for your suggestion. We have removed the patient data from the manuscript.

  1. The references cited in this manuscript are appropriate and relevant to this research.

Response: Thank you for your positive feedback regarding the references cited in our manuscript.

  1. Overall, the presented results (animal model) can support the conclusion, but this manuscript still needs a revision before it can be accepted.

Response: Thank you once again for your thoughtful evaluation of our manuscript and your valuable feedback. We have carefully addressed your comments and revised the manuscript accordingly. We hope that the revised manuscript addresses your concerns and meets the criteria for acceptance.

Reviewer 2 Report

Comments and Suggestions for Authors

The authors set out to investigate the possibility of improving liver transplantation by precooling, clearly realizing that this method is more than 50 years old and should be done something novel. Still, there are inconsistencies in the quality of grafts taken by two different techniques. The work is well done and may interest readers of Biomedicine. However, there are a few points that need to be corrected.

Taking into account the authors' phrase..... However, outcomes from DCD transplants have historically shown a relative inferiority when compared to liver transplantation from a living donor or donors after brain death [9,10]… it should be clearly stated that the authors are modeling this very situation (namely DCD, don’t they?).

….rotenone + antimycin A (2 μM final)….. 2 μM final of what? Rot? Or AntA? Or both by 2 μM?

The authors show that the activity of complex 1 drops by a factor of 2.5 (from 25 to 10). At the same time in SeaHorse experiments, this should be reflected on the starting point, and there the difference is from about 12 to 20, i.e. the difference is significantly less than 2-fold. The authors should, first of all, say what respiratory substrate they used and somehow explain this difference, which is small, but somewhere there is a reason for it. Moreover, if we calculate the respiratory control, by eye ball it seems that all three curves show the same respiratory control, i.e. the oxidative phosphorylation system is not damaged, but simply the number of working mitochondria decreases.  It also should be written somewhere and I wonder how it could be since oxidative phosphorylation is the most sensitive to damage factor.

Fig 3 C. What is state 4o? (Maybe state 4?). What is state 3μ? (Maybe state 3u?)

Author Response

We sincerely appreciate all valuable comments and suggestions from the reviewers and editors, which helped us improve the quality of our manuscript. We have thoroughly revised the manuscript as the reviewers suggested. We believe that all the reviewers’ and editor’s concerns have been carefully addressed. Our point-by-point response to the reviewers' comments is provided below in blue.

Review 2. The authors set out to investigate the possibility of improving liver transplantation by precooling, clearly realizing that this method is more than 50 years old and should be done something novel. Still, there are inconsistencies in the quality of grafts taken by two different techniques. The work is well done and may interest readers of Biomedicine. However, there are a few points that need to be corrected.

Response: Thank you for your insightful feedback and for recognizing the potential impact of our work. We are grateful for your positive assessment of our work and believe that the corrections and clarifications we made below and incorporated into the main manuscript will further strengthen the manuscript. We hope that the improved manuscript will meet the standards of Biomedicine's readership.

Taking into account the authors' phrase..... However, outcomes from DCD transplants have historically shown a relative inferiority when compared to liver transplantation from a living donor or donors after brain death [9,10]… it should be clearly stated that the authors are modeling this very situation (namely DCD, don’t they?).

Response: Thank you for pointing this out. We have deleted this sentence from the manuscript as we did not include the DCD in our study.

….rotenone + antimycin A (2 μM final)….. 2 μM final of what? Rot? Or AntA? Or both by 2 μM?

Response: We meant that the final concentrations for both rotenone and antimycin are 2µM. We have clarified this in the main manuscript as “rotenone + antimycin A, each at a final concentration of 2 µM.”

The authors show that the activity of complex 1 drops by a factor of 2.5 (from 25 to 10). At the same time in SeaHorse experiments, this should be reflected on the starting point, and there the difference is from about 12 to 20, i.e. the difference is significantly less than 2-fold. The authors should, first of all, say what respiratory substrate they used and somehow explain this difference, which is small, but somewhere there is a reason for it. Moreover, if we calculate the respiratory control, by eye ball it seems that all three curves show the same respiratory control, i.e. the oxidative phosphorylation system is not damaged, but simply the number of working mitochondria decreases.  It also should be written somewhere and I wonder how it could be since oxidative phosphorylation is the most sensitive to damage factor.

Response: Thank you for raising this important question. We have clarified the substance used in the OCR measurement in the method section and here: Two micrograms of freshly prepared mitochondria in mitochondrial assay solution (MAS) (70 mM sucrose, 220 mM mannitol, 10 mM KH2PO4, 5 mM MgCl2, 2 mM HEPES, 1 mM EGTA, and 0.2% BSA in nanopure water, pH 7.4, adjusted using KOH) were loaded onto the 24-well SeaHorse plates. Succinate (10 mM) and Rotenone (2 μM) were added to the MAS and the resulting solution was used to make 10 × stocks of the respiratory inhibitors and uncoupler. The substance of Succ/Rot (succinate with rotenone) bypasses complex I to feed electrons directly to the ubiquinone pool. There are no significant differences between the precooling and sham groups in state 2 before the addition of ADP, but significant differences exist between the precooling and control groups (Fig. 3C). This demonstrates that oxidative phosphorylation is more damaged in the control group than in the precooling group.

Fig 3 C. What is state 4o? (Maybe state 4?). What is state 3μ? (Maybe state 3u?)

Response: State 4o means that the ATP synthase is inhibited with the addition of oligomycin (State 4o). We added the explanation in the main text. We apologize for the typo in "state3µ." We have corrected it in the main text to "state 3u."

Reviewer 3 Report

Comments and Suggestions for Authors

In this study, the authors tried to investigate whether inducing local hypothermia in the liver before the stopping of blood flow protects against IRI. The authors concluded that local precooling attenuated oxidation stress and inflammatory response in the liver, ameliorated hepato-cellular damage and improved transplanted liver graft function.

Comments:

This is an interesting study. The reviewer has some comments as follows:

1.     This manuscript is well written. After searching the literature, this study has its novelty.

2.     Basically, the experimental design of the animal model in this study is reasonable.

3.     In the Methods section, the fonts of the subtitles are inconsistent, please check the fonts.

4.     In line 277 and Figure 3C, please check the words “state 3μ” (state 3u).

5.     In Figure 4D, please check the scale of image for caspase-3 IHC in OLT-Control group. The cells in this image appear to be much smaller than those in the sham control group.

6.     In Table 1, please confirm whether the font is consistent with the manuscript.

7.     In Figure 6, the data presentation for donor body temperature during liver procurement surgery correlates with transplanted graft outcomes is confusing. The conditions of these patient data are very different from the animal data. The temperature changes only 34-38oC and there are no precooling or cold processes in patient data. These patient data cannot echo the results of animal experiments. Therefore, please consider whether the presentation of these patient data is necessary.

8.     The references cited in this manuscript are appropriate and relevant to this research.

9. Overall, the presented results (animal model) can support the conclusion, but this manuscript still needs a revision before it can be accepted.

Author Response

We sincerely appreciate all valuable comments and suggestions from the reviewers and editors, which helped us improve the quality of our manuscript. We have thoroughly revised the manuscript as the reviewers suggested. We believe that all the reviewers’ and editor’s concerns have been carefully addressed. Our point-by-point response to the reviewers' comments is provided below in blue.

Review 2. The authors set out to investigate the possibility of improving liver transplantation by precooling, clearly realizing that this method is more than 50 years old and should be done something novel. Still, there are inconsistencies in the quality of grafts taken by two different techniques. The work is well done and may interest readers of Biomedicine. However, there are a few points that need to be corrected.

Response: Thank you for your insightful feedback and for recognizing the potential impact of our work. We are grateful for your positive assessment of our work and believe that the corrections and clarifications we made below and incorporated into the main manuscript will further strengthen the manuscript. We hope that the improved manuscript will meet the standards of Biomedicine's readership.

Taking into account the authors' phrase..... However, outcomes from DCD transplants have historically shown a relative inferiority when compared to liver transplantation from a living donor or donors after brain death [9,10]… it should be clearly stated that the authors are modeling this very situation (namely DCD, don’t they?).

Response: Thank you for pointing this out. We have deleted this sentence from the manuscript as we did not include the DCD in our study.

….rotenone + antimycin A (2 μM final)….. 2 μM final of what? Rot? Or AntA? Or both by 2 μM?

Response: We meant that the final concentrations for both rotenone and antimycin are 2µM. We have clarified this in the main manuscript as “rotenone + antimycin A, each at a final concentration of 2 µM.”

The authors show that the activity of complex 1 drops by a factor of 2.5 (from 25 to 10). At the same time in SeaHorse experiments, this should be reflected on the starting point, and there the difference is from about 12 to 20, i.e. the difference is significantly less than 2-fold. The authors should, first of all, say what respiratory substrate they used and somehow explain this difference, which is small, but somewhere there is a reason for it. Moreover, if we calculate the respiratory control, by eye ball it seems that all three curves show the same respiratory control, i.e. the oxidative phosphorylation system is not damaged, but simply the number of working mitochondria decreases.  It also should be written somewhere and I wonder how it could be since oxidative phosphorylation is the most sensitive to damage factor.

Response: Thank you for raising this important question. We have clarified the substance used in the OCR measurement in the method section and here: Two micrograms of freshly prepared mitochondria in mitochondrial assay solution (MAS) (70 mM sucrose, 220 mM mannitol, 10 mM KH2PO4, 5 mM MgCl2, 2 mM HEPES, 1 mM EGTA, and 0.2% BSA in nanopure water, pH 7.4, adjusted using KOH) were loaded onto the 24-well SeaHorse plates. Succinate (10 mM) and Rotenone (2 μM) were added to the MAS and the resulting solution was used to make 10 × stocks of the respiratory inhibitors and uncoupler. The substance of Succ/Rot (succinate with rotenone) bypasses complex I to feed electrons directly to the ubiquinone pool. There are no significant differences between the precooling and sham groups in state 2 before the addition of ADP, but significant differences exist between the precooling and control groups (Fig. 3C). This demonstrates that oxidative phosphorylation is more damaged in the control group than in the precooling group.

Fig 3 C. What is state 4o? (Maybe state 4?). What is state 3μ? (Maybe state 3u?)

Response: State 4o means that the ATP synthase is inhibited with the addition of oligomycin (State 4o). We added the explanation in the main text. We apologize for the typo in "state3µ." We have corrected it in the main text to "state 3u."

Review 3.

This is an interesting study. The reviewer has some comments as follows:

  1. This manuscript is well written. After searching the literature, this study has its novelty.

Response: Thank you for your positive feedback and for recognizing the novelty of our study.

  1. Basically, the experimental design of the animal model in this study is reasonable.

Response: We really appreciate your positive evaluation of our experimental design.

  1. In the Methods section, the fonts of the subtitles are inconsistent, please check the fonts.

Response: Thank the reviewer for pointing this out. We have carefully reviewed and standardized all the fonts.

  1. In line 277 and Figure 3C, please check the words “state 3μ” (state 3u).

Response: we apologize for the typo. We have corrected it to state 3u.

  1. In Figure 4D, please check the scale of image for caspase-3 IHC in OLT-Control group. The cells in this image appear to be much smaller than those in the sham control group.

Response: We have replaced the image with another from the same group and confirmed the scales.

  1. In Table 1, please confirm whether the font is consistent with the manuscript.

Response: We have updated the fonts and ensured they are consistent with the rest of the manuscript.

  1. In Figure 6, the data presentation for donor body temperature during liver procurement surgery correlates with transplanted graft outcomes is confusing. The conditions of these patient data are very different from the animal data. The temperature changes only 34-38oC and there are no precooling or cold processes in patient data. These patient data cannot echo the results of animal experiments. Therefore, please consider whether the presentation of these patient data is necessary.

Response: Thank you for your suggestion. We have removed the patient data from the manuscript.

  1. The references cited in this manuscript are appropriate and relevant to this research.

Response: Thank you for your positive feedback regarding the references cited in our manuscript.

  1. Overall, the presented results (animal model) can support the conclusion, but this manuscript still needs a revision before it can be accepted.

Response: Thank you once again for your thoughtful evaluation of our manuscript and your valuable feedback. We have carefully addressed your comments and revised the manuscript accordingly. We hope that the revised manuscript addresses your concerns and meets the criteria for acceptance.

Round 2

Reviewer 3 Report

Comments and Suggestions for Authors

This revised manuscript has a great improvement. No further comments.